

# A large-scale wind turbine model installed on a floating structure: experimental validation of the numerical design

Federico Taruffi[1], Simone Di Carlo[1], Sara Muggiasca[1], Marco Belloli[1]

[1]Mechanical Engineering Department, Politecnico di Milano, Milano, 20156, Italy

*Correspondence to*: Federico Taruffi (federico.taruffi@polimi.it)

**Abstract.** In the field of floating wind energy large-scale wind turbine models deployed in natural environment represent a key link between small-scale laboratory tests and full-scale prototypes. While implying smaller cost, design and installation effort than a full-scale prototype, large-scale models are technologically very similar to prototypes, can be tested in natural sea and wind conditions and reduce by a consistent amount the dimensional

scaling issues arising in small-scale experiments. In this framework the presented work is reporting the aerodynamic and control system assessment of a 1:15 model of the DTU 10 MW wind turbine installed on a multipurpose platform model for fish farming and energy production. The model has operated for six months in a natural laboratory and has been exposed to fully natural and uncontrolled environmental conditions. Assessment is performed in terms of rotor thrust force and power controller parameters such as rotor speed, blade pitch and

rotor power as a function of incoming wind speed.

## 1.    Introduction

Nowadays the continuously increasing demand for green energy production and the quest for sustainable food is pushing forward the exploitation of ocean areas for resources that it is more and more difficult to harness on land (Jouffrey, et al., 2020). Renewable ocean energy production technologies are mainly focused on wave energy,

tidal energy or floating wind; on the other side, as far as food production is concerned, fish farming is the leading activity. It is known, regarding this last statement, that fish farming activity are concentrated along shores for ease of logistics and to benefit from more moderate sea conditions, and effort is required to install farms in more open waters.

In the framework of ocean areas exploitation a novel technology now under research are multipurpose platforms

(Abhinav, et al., 2020). A multipurpose platform is a floating platform hosting different technologies for contemporary energy and food production. In this way different activities can improve their redditivity by sharing common and expensive facilities, like the platform itself, the mooring system, the electrical dispatch system and so on. An example in this sense is the multipurpose platform designed during the H2020 project "The Blue Growth Farm" (Lagasco, et al., 2019), and whose model scale tests are the main topic of this paper. The full-scale platform

is a barge-shaped floater hosting a moonpool for fish farming, that is the primary activity to be performed on the platform; then, wave energy converters are exploiting wave motions and platform motions for energy production. Last, a multi-megawatt wind turbine is exploiting wind power. It is obvious to think that from a design point of view it is extremely difficult to numerically model such a system, given the contemporary occurrence of several subsystems, each one characterized by its own way of operation, and by the existence of several fields of

engineering all together, like hydrodynamics, aerodynamics, turbomachinery, structural dynamics.



It becomes way important, then, to be able to perform experiments on scaled models. The aim of model tests is to inspect aspects of the system that are not visible in numerical modelling, and at the same time to validate or calibrate numerical models. During the aforementioned project, actually, two test campaigns were performed to investigate the behaviour of the platform and get hints on its feasibility. The first campaign was about a 1:40 model tested in a wave tank, at ECN; the second campaign, explained in this paper, was conducted on a 1:15 model, then a large-scale model, to inspect the behaviour of the platform in open sea (Ruzzo, et al., 2021). Large-scale models deployed in a natural outdoor environment are a valid complement to traditional small-scale laboratory models in understanding the real features of the system and in updating or validating codes. Large-scale models allow to increase the fidelity of the experiment by reducing the scaling effect: focusing on wind turbines, adopting a large scale allows to better reproduce the aerodynamic behaviour of the rotor, that in laboratory scale experiments is usually impaired by low-Reynolds effects. Another advantage in large-scale modelling is represented by the chance to operate in a natural environment, so exposed to uncontrolled environmental actions, further enhancing the fidelity of the experiment. Moreover, such models can be built to be very similar to full-scale prototypes from the point of view of construction, technology involved and operation, allowing to reach higher TRLs (Technology Readiness Level), about 5-7.

Actually, the platform was moored for ten months in the waters of the Messina's strait, in front of the city of Reggio Calabria, and the experimental campaign was targeted to the inspection of feasibility of the whole system as a combined food and energy production platform. The present work is focused particularly on the assessment of the aerodynamic and controller performances of a 1:15 wind turbine model of the 10 MW DTU reference wind turbine (Bak, et al., 2013) that is installed on the platform. It is of interest to understand if the performances in terms of power curve and thrust curve are respecting the operational parameters that were established during the design phase. This operation is performed by inspecting the experimental behaviour of the wind turbine model in particular, steady state operation points and by comparing relevant parameters to the ones found in the numerical simulations.

The paper is structured as follows: Sect. 2 is explaining briefly the characteristics of the multipurpose platform scaled model and of the wind turbine model, Sect. 3 is detailing sensor setup and test procedures, Sect. 4 is reporting some preliminary checks to verify the consistency of gathered data, Sect. 5 is detailing the assessment of aerodynamic performances, Sect. 7 is concluding the work.

## 2. The large-scale model

The large-scale model of the platform is a rectangular barge with sides equal to 14 and 10.8 m, with a draft of 2 m. One of the short sides of the platform is hosting a row of wave energy converters of the REWEC type that can be opened or closed depending on the needs of the campaign. The wave energy converters are not equipped with any energy conversion device, and a calibrated hole is reproducing the correct air-flow condition (Thiebaut, Pascal, & Andreu, 2015). On the same side of the platform it is located a 1:15 scaled model of the 10 MW DTU wind turbine, with the aim of recreating the dynamical effect of the wind turbine on the full-scale platform. Then, the centre of the platform is hosting a moonpool for fish farming; the fish cages are in this way shielded by waves (Figure 1).



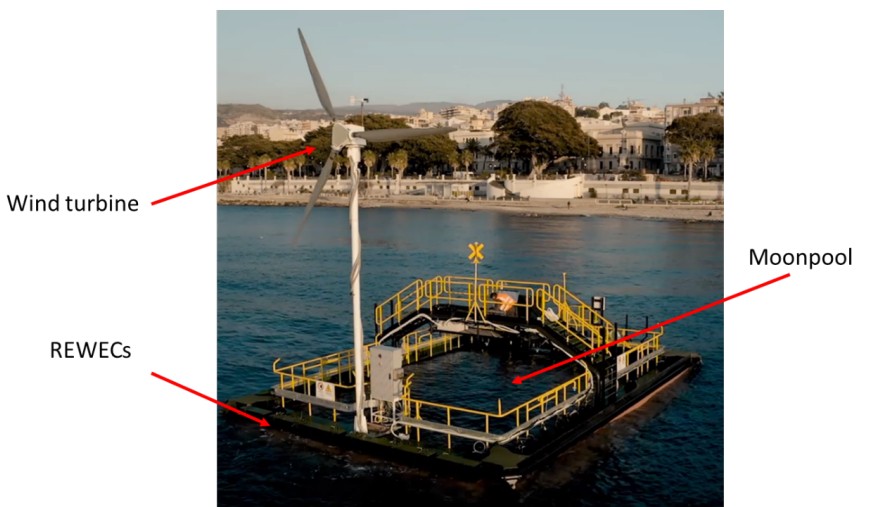

**Figure 1: Overview of the model and principal subsystems**

### 2.1. Wind turbine model

The wind turbine is a 1:15 scaled model of the 10 MW DTU reference wind turbine, designed to be an aeroelastic
model and to reproduce the effect of a full-scale wind turbine (Muggiasca S. , et al., 2021).

The design reflects the necessity to build a scale model and a real, fully functioning machine to be operated in an
outdoor and not-confined environment at the same time. For this reason, the safety issues were considered of
utmost importance. The full-scale reference was scaled follows a performance scaling approach: the goal is to

reproduce the 1:15 scaled thrust force. A hybrid scale law was adopted, allowing to obtain the same goal
performances but with a smaller rotor (Fontanella, Taruffi, Muggiasca, & Belloli, 2019). For the blade airfoil the
SG6040 was selected, as it is suitable for low-Reynolds applications, and it was experimentally characterized by
means of wind tunnel tests on a 2D model. The rotor aerodynamics was designed following an iterative procedure
obtaining the chord and twist distribution along the blades, as detailed in (Muggiasca S. , Taruffi, Fontanella, Di

Carlo, & Belloli, 2021).

To grant the structural integrity even under extreme wind and wave events, a structural assessment was performed.
All the crucial components, including the tower and rotor-nacelle assembly, were verified by means of FEM
analysis. The GFRP composite blade layup was verified by means of static experimental tests performed on a
blade prototype. The tower aeroelastic constraint was fulfilled with the resulting first natural frequency matching

the target, while for the blades the safety concerns were found to be primary (Muggiasca S. , Taruffi, Fontanella,
Di Carlo, & Belloli, 2021).

The turbine has five degrees of freedom: the rotor rotation, three individual drives for blade pitch and yawing of
the nacelle. An embedded control and monitoring system supervises the turbine to ensure the full operation of the
machine, similarly to the full-scale one, the management of the status and the signals monitoring and acquisition.

In particular, the control system is a derivation of the variable-speed variable-pitch algorithm developed for the





DTU 10 MW (Bak, et al., 2013). It features a startup procedure, partial- and full-load operation and shutdown action (Muggiasca, et al., 2019).

Relevant data about the model are listed in Table 1 and the complete design can be found in (Muggiasca S. , et al., 2021).

**Table 1: Gross data about the wind turbine model**

| Model scale [-] | 1:15 |
|---|---|
| Number of blades [-] | 3 |
| Rotor diameter [m] | 6.9 |
| Blade length [m] | 3.1 |
| Hub height above SWL [m] | 8 |
| Cut-in wind speed [m/s] | 1.8 |
| Rated wind speed [m/s] | 5 |
| Cut-out wind speed [m/s] | 11 |
| Rated rotor speed [rpm] | 110 |
| Rated thrust force [N] | 479 |
| Rated power [W] | 1328 |

## 3. Experimental campaign

The floating multipurpose platform hosting the wind turbine large-scale model was deployed in the end of February 2021 in the waters of Messina's strait, in front of the city of Reggio Calabria, in southern Italy. In this area, the Natural Ocean Engineering Laboratory (NOEL) (Arena & Barbaro, 2013) (NOEL - University of Reggio

Calabria, 2022) is performing experiments in outdoor conditions on several ocean engineering applications. Actually, the coexistence of peculiar wind conditions, wind fetch and sea current causes wave spectra to be a scaled version of oceanic wave spectra, then making this place suited to be a natural laboratory for offshore engineering scaled experiments, that were conducted here also in past times. The experimental campaign then started in February 2021 and the turbine was operated from April 2021 to July 2021 (Ruzzo, et al., 2021). Photos

of the multipurpose platform on the site of deployment and of the large wind turbine scale model are shown in Figure 2 and Figure 3. General scope of the tests is to prove the feasibility of the concept as a multipurpose offshore system for aquaculture, wave energy production and wind energy production; more in detail it is interesting to evaluate how much these subsystems are interfering one with the other, and if this interference is detrimental for their efficient operation. Moreover, these large-scale experiments represent a sort of intermediate

step between small scale model tests performed in the wave basin and the full-scale one. While in a wave basin just the preliminary assessment of the dynamic behaviour of the prototype can be achieved in a fully controlled setting, in this experiment the behaviour and the feasibility of the concept can be inspected from several points of view, thanks to the greater dimensions of the model and to the exposure to the natural environments. Greater dimensions allow to have a state-of-the art structural monitoring system, electrical dispatch system, realistic

operation of integrated technologies; the outdoor conditions guarantees of course the presence of natural sea and wind conditions, including extreme events, the occurrence of corrosion, wear and marine growth. From the wind turbine side, instead, the aim of the experimental campaign is to evaluate the influence of floating conditions (e.g. platform motions) on turbine power production and turbine structural health.



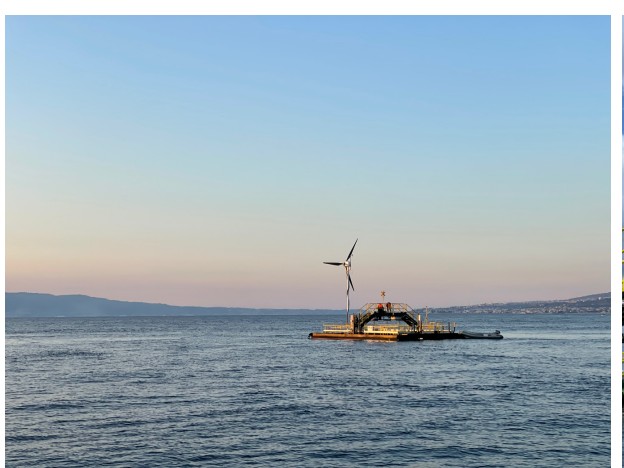 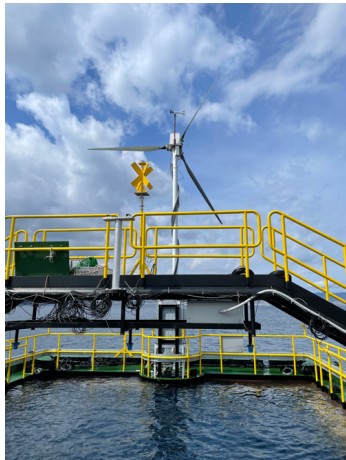

**Figure 2: The multipurpose platform**

**Figure 3: The large-scale wind turbine model**

### 3.1. Sensors setup

For the sake of aerodynamic validation, structural monitoring and more in general for scientific investigation of floating wind turbine operation, the model is equipped with a system of sensors devoted to on demand data gathering during the whole experimental campaign. Firstly, a propeller anemometer placed on the top of the nacelle is measuring the wind speed and direction, relative to the yaw angle of the nacelle itself. Rotor main drive and blade pitch drives encoders are registering the actual rotor speed, rotor angle and blade pitch angle; a current sensor on the main generator is used to evaluate the torque on the generator side, and consequently the generator power. Data gathered by these sensors allow in detail to evaluate the Cp-λ curves of the rotor (Cp is the power coefficient and λ or TSR is the tip speed ratio). Derived quantities of mechanical power, Cp and TSR are calculated from the measurements as follows:

$$P = \Omega_G \cdot Q_G \qquad C_P = \frac{P}{\frac{1}{2} \cdot \rho \cdot A \cdot U^3} \qquad TSR = \frac{\Omega_R \cdot r}{U} \qquad \textbf{(1)}$$

where $\Omega$ is the rotational speed (reported either at rotor or generator side), $Q$ is the generator torque, $U$ is the wind speed, $\rho$ is the standard air density, $r$ is the rotor radius and $A$ is the rotor area.

A summary of measurements for wind turbine power capabilities is reported in Table 2

**Table 2: List of signals to assess rotor power performances**

| Signal | Unit | Sensor |
|---|---|---|
| Wind speed | m/s | Propeller anemometer |
| Wind direction | deg | Propeller anemometer |
| Rotor speed | rpm | Main drive encoder |
| Rotor position | deg | Main drive encoder |
| Rotor torque | Nm | Main drive current sensor |

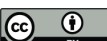



| Blade pitch | deg | Pitch drive encoder |
|---|---|---|

Sensors set is completed by a structural monitoring system composed by a set of strain gauges on tower base, a set of strain gauges on the blades and some accelerometers; this set of sensors is instead devoted to loads and vibrations monitoring. Tower base strain gauges are arranged in two half-bridge configurations, to measure flexional stresses on two perpendicular axes, and a full bridge for torsional stress sensing. Blade strain gauges are arranged so as to measure flapwise and edgewise bending stresses and torsional stresses on one of the blade, being

the other two provided only with flapwise bending stresses measure. Accelerometric measures are accomplished by means of two triaxial accelerometers, one placed in the nacelle and the other at tower-mid (Figure 4).

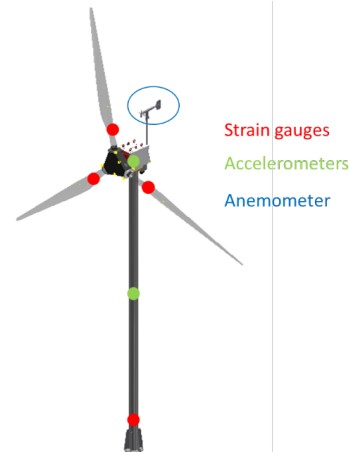

**Figure 4: Anemometer, accelerometers and strain gauges locations**

Data are acquired by a NI-PXIe at a 50Hz sampling rate. Files containing time histories are sent to a laboratory

on the shore and saved in a cloud.

### 3.2. Wind turbine operation

The wind turbine is normally kept parked, and it is operated when sea conditions are favourable, so to avoid dangerous operation. Rotor is first oriented towards the wind direction and then put into rotation under the

supervision of the user. Data acquisition is active during each operation, then a typical time history comprises a startup phase, with the main motor accelerating the rotor, an operating phase, when rotor speed and torque are regulated by the VSVP controller, and a shutdown phase, when the blades are feathered to 90 deg and the rotor is stopped and then braked. Depending on wind conditions, the wind turbine is setting to an operating point defined by rotor speed, torque and blade pitch; among all operating points a major division is made of course in below-

rated conditions and above-rated conditions, with different control system behaviour. The length of a time history is usually 30 min or 1-h so to have a sufficiently long time duration of continuous operation data.

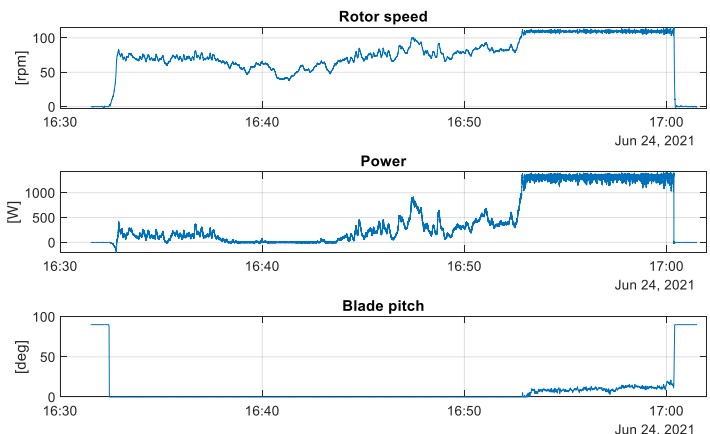

**Figure 5: An example of time history with both a below- and an above-rated regime**

**4. Preliminary data analysis**

**4.1. Inspection of tower dynamics**

As one of the first tasks of the experimental campaign, the dynamic behaviour of the tower is investigated. Due to tower dimensions, it was not possible to perform a complete modal analysis of the system, then an on-site investigation is preferred. This step is necessary to verify the FE tower numerical model used during the design
phase and to have an insight into tower damping, a quantity that is difficult to evaluate by theory. To obtain this piece of information a decay test is performed: starting from a rated operation condition, with maximum value of thrust, the blades are abruptly feathered so to trigger a free-decay of the tower. Free decay acceleration signal (Figure 6) is then acquired and post-processed.

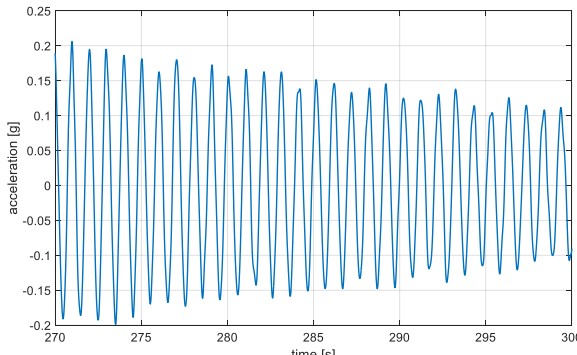

**Figure 6: Free decay as measured by nacelle accelerometers**

Preliminary results of the numerical model of the tower individuated the first tower mode at 1.05 Hz and the second at 14 Hz. Being the second mode of vibration out of the expected harmonic forcing, major attention is put





on the experimental verification of the first mode frequency. The decay signal of course contains also higher modes contribution, even if in a slight manner, and some low frequency content due to slow platform motions; application of a bandpass filter allows to obtain the 1-dof decay corresponding to the first mode of vibration (Figure 7). Analysis of the signal revealed a first mode frequency of 0.98 Hz and a damping ratio of 0.33 %, measured by the logarithmic decrement procedure. Besides this, given the presence of two accelerometers along the tower, an approximate verification of the mode shape was also possible by comparing the numerical mode shape to the amplitudes of oscillation at tower top and tower mid. This verification resulted in a 3% mismatch on first mode shape (Figure 8). The free-decay signal, this time not low-pass filtered, gave also an insight on the second tower mode, located around 12 Hz. As can be seen, experimental natural frequencies are lower than the numerically predicted ones, and some reason for this mismatch can be here inferred. Firstly, in the FEM model the constraint at the tower base is modelled as a perfect clamp, while in reality a bolted joint is connecting the tower base flange to the steel hull of the platform, allowing some flexibility; secondly, some more mass should be accounted for on the nacelle and on the tower, given by cable bundles, sheats, bolting, paint.

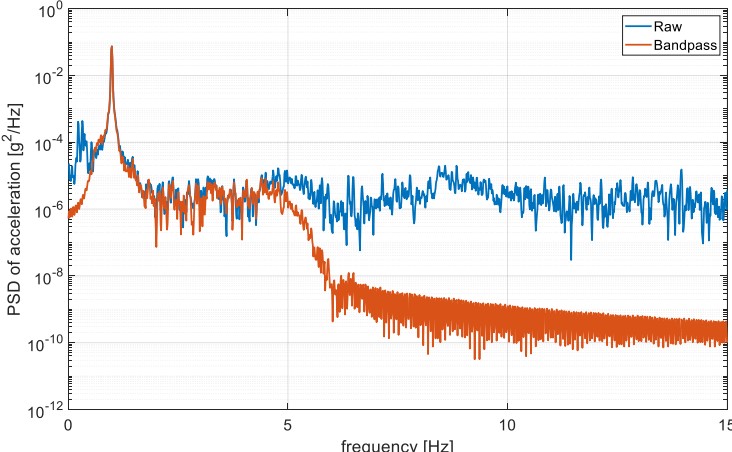

**Figure 7: PSD of raw or bandpass-filtered decay signal**

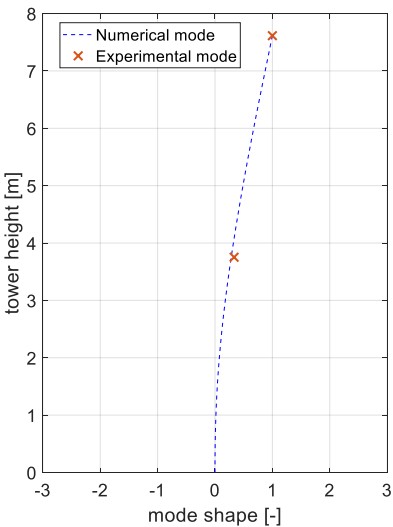

**Figure 8: Experimental and numerical modal shapes**

## 4.2. Wind correction

To correctly evaluate wind turbine performance curves, in terms of thrust and power, a reliable measurement of wind is necessary. The large-scale model is indeed equipped with a propeller anemometer placed on the nacelle, at a height of approximately 1 m above the tower-nacelle connection. This sensor is providing wind speed and wind misalignment with respect to yaw heading. It is obvious that the wind measurement obtained when the wind turbine is spinning is influenced by the shade of the rotor, given that the anemometer is always oriented in the direction of incoming wind, as the rotor itself is. The influence is observed to be more or less intense as the rotor is spinning with different velocities: it is found generally that the influence of the rotor wake is more intense when the spinning velocity is low, then in below-rated operation, leading to an underestimation of the measure; as the rotor speed is increased, instead, the wind speed measurement is less and less influenced. When the wind turbine is parked, instead, the wind measurement obtained with this sensor is deemed to be reliable. To obtain a wind speed measure that is as much as possible reliable when the rotor is spinning, a procedure to correct wind measurement has been formulated, and it is hereby presented. The procedure takes into advantage the presence of an onshore anemometer of sonic type, installed on a 5 m pole on the shore facing the platform. As the wind turbine anemometer is considered the most reliable wind speed measure when the wind turbine is parked, mainly because it is located at hub height and close to the rotor thus measuring the actual wind that hits the turbine, firstly a correction coefficient is derived between the offshore and the onshore anemometer in non-spinning conditions. The correction coefficient is of course variable both in speed and in direction, because the onshore anemometer reading is influenced by some obstacles present nearby; nevertheless, the variability in direction was found to be negligible. A second coefficient is derived between the onshore and the offshore anemometer for spinning cases (i.e. the measure to be corrected), always considering the wind speed variability (that includes the rotor speed variability, given the unique operating points of the machine). The two coefficients are then merged obtaining a



single coefficient to be multiplied to offshore anemometer measure in spinning rotor conditions to evaluate the
correct measure and overcome the shadow effect.

The correction procedure is summed up in Eq. (2) where $U$ stands for wind speed, $ns$ for non-spinning condition, $s$ for spinning condition, $off$ for offshore, $on$ for onshore and $CORR$ for corrected measure; the coefficients are function of the wind speed. First the coefficients $c_{ns}$, $c_s$ and $C$ are evaluated and then the corrected measure is obtained by applying the coefficient $C$ on the original, offshore, measure.

$$c_{ns}(U) = \frac{U_{on}^{ns}}{U_{off}^{ns}} \qquad c_s(U) = \frac{U_{on}^s}{U_{off}^s} \qquad C(U) = \frac{c_s(U)}{c_{ns}(U)} \qquad U_{off\,CORR}^s = U_{off}^s \cdot C(U) \qquad \textbf{(2)}$$


To calculate the coefficients in the range 2.5 to 9.5 m/s, two curve fitting procedures were performed separately for the points collected in below-rated and above-rated regions, because the coefficients clearly present a different trend. This fact is considered reasonable because in below-rated the rotor spins at different and increasing speeds while in above-rated the rotor speed is constant. In the below-rated region a decreasing trend is observed and the
obtained coefficient is $C(U) = -0.16U + 1.78$, while in above-rated region the trend is constant and approximately equal to 1, thus no correction is applied to the measures for wind speeds greater than 5 m/s.

The correction here presented is applied to all the wind speed data in spinning rotor conditions utilized in this work, while no correction is applied for non-spinning cases. As for the wind direction, the offshore measure is considered reliable also in spinning conditions.


### 4.3. Operating points

In view of the validation of the design, the wind turbine operational parameters should be evaluated so that the extracted values are meaningful. It is necessary, then, to extract from time histories, time windows of data with peculiar characteristics, here termed operating points. We can define an operating point as a time interval in which
all the environmental conditions and wind turbine parameters are constant, meaning that the machine is set to a steady state point, or regime point. It is obvious that pure constant conditions on parameters are not existing, but nevertheless, this fact is not impeding to find a regime point with sufficiently stationary statistical characteristics. Each point is characterized by a given wind speed, rotor speed and blade pitch. In order to individuate operating points the following procedure was considered. All the time history records featuring the wind turbine in working
condition (identified inquiring rotor speed greater than 60 rpm, commanded torque greater than 0 and collective blade pitch less than 30 deg) were cut into 10-seconds frames and the average value of measures calculated on each selected frame are considered as operating points. For each frame, the operating conditions are considered stationarity if the rotor speed variance is within a certain range, discarding transient conditions that can alter the resulting performance. The alignment of the rotor with the wind direction is checked too, discarding frames with
greater misalignment that can show performances altered with respect to the ideal, fully aligned case.

The obtained operating points are then utilized to evaluate the machine performances and calculate the thrust force and power coefficients for the purpose of validating the design. The operating points are further averaged considering 0.25 m/s wide wind speed ranges spacing from 2.5 to 9.5 m/s in order to obtain a curve as function of wind speed.

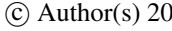


Considering the randomness of the environmental conditions, months of testing are needed to collect a sufficiently high number of regime points at several wind speeds, covering all the working regions of the VSVP controller.

## 5. Aerodynamic design validation

For the good result of the experimental campaign, that aims at reproducing at model scale the MPP concept and all its subsystems, it is of fundamental importance to assess that the wind turbine model behaves as it was laid out in the design phase. In particular, this refers to the aerodynamic loads and the performances of the turbine, pictured by the thrust force (the aerodynamic force acting perpendicular to the rotor plane and considered crucial in particular for FOWT dynamics) and the power output. Thus it is necessary to validate the aerodynamic design to ensure the quality of the results. This is accomplished by comparing the numerical power coefficient curve with

experimental values and by matching the numerical and experimental thrust curves. The comparator numerical data were the base of the wind turbine design and were in turn assessed with the targets given by the full-scale concept (Muggiasca S. , et al., 2021).

### 5.1. Power coefficient

The power output of the wind turbine is representative of the aerodynamic performances. The power output, nondimensionalized in terms of power coefficient (Cp), is the term of comparison. The numerical Cp curves, expressed as function of tip speed ratio (TSR or λ) and discretized in pitch angle, were evaluated by means steady-state simulation in Fast v8 (Jonkman & Buhl, 2005) for the aerodynamic design of the model and they were successfully matched with the Cp curved of the full-scale reference (Muggiasca S. , Taruffi, Fontanella, Di Carlo,

& Belloli, 2021). The experimental Cp values are calculated for the steady-state operating points identified as in Sect. 4.3. The comparison is shown in Figure 9: experimental measurements (dots) are compared with numerical curves evaluated in FASTv8 (lines). It focuses on the above-rated region: the modifications introduced in the torque controller (see Sect 6.1) that results in a non-optimal Cp tracking in the region, the nature of the graph itself (all points would nearly collapse in one) and the higher uncertainty on the speed measure make a comparison of

the values in the below-rated region less significative. A good match between experimental measurements and target can be found: the experimental points lay on the correct Cp-λ for the same pitch angle. Thus, the aerodynamic performances of the rotor meet the expectation and correctly reproduces the numerical design.

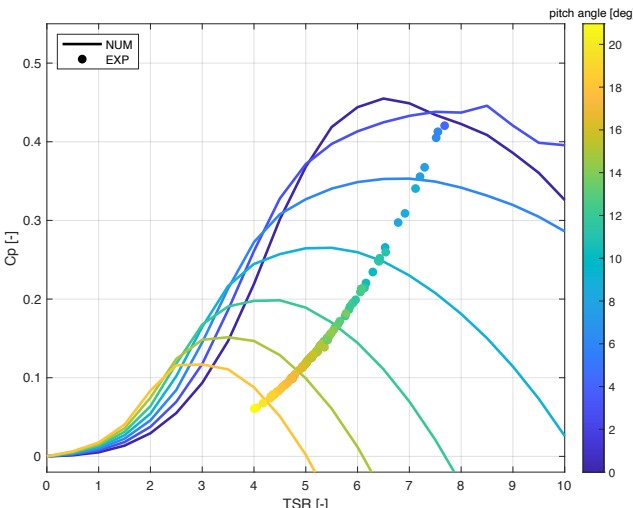

**Figure 9: Power coefficient (Cp) as function of TSR and pitch angle**

### 5.2. Thrust force evaluation

Thrust measurement can simply be accomplished by placing a load cell under the nacelle, and this is actually the procedure followed in small-scale models for wind tunnel experiments (Bayati, Facchineti, Fontanella, Taruffi, & Belloli, 2020). In the present case, however, several factors related to the model construction and the experimental campaign characteristics made impossible the use of a load cell on the model. A load cell would not have guaranteed the necessary mechanical resistance and stiffness at the connection between tower and nacelle, and in addition to this, the accommodation of the yaw mechanism and cable routes descending from the nacelle would have been too complicated from the constructional point of view. Moreover, the load cell protection from environmental actions like rain, salty water would not have been a straightforward task. For these reasons it has been decided during the model design phase to evaluate the thrust force on the rotor in an indirect way, then by measuring the bending deformation on the tower, exploiting the strain gauges sets. In addition, the thrust was indirectly evaluated measuring the deformation on the blades root, and the results were compared.

#### 5.2.1. Tower strain gauges calibration

Due to the dimensions of the tower and the peculiarity of its installation, it was not possible to calibrate the strain gauges with the same acquisition system used during the experimental campaign, and then the output of strain gauges resulted biased by an offset from zero and a multiplicative coefficient. Offset was obtained by acquiring strain gauges signal in calm wind and sea conditions, corresponding to a practically zero bending moment on the tower base. For the multiplicative coefficient, instead, a more sophisticated procedure has been performed, exploiting the accelerometer system. The starting point of the procedure is the free-decay signal obtained during the dynamic investigation of the tower (Figure 6). Resulting accelerometric signal has been low-pass filtered to isolate the first mode of the pole and the 1-dof acceleration decay is then integrated to get nacelle displacement time history.

The measured displacement time history is coupled to the first modal shape of the tower, obtained with a FEM software. The mode shape $w(x)$ is fitted with a $6^{th}$ order polynomial as in (3), then following Bernoulli's beam



theory the strain along the beam can be obtained with Eq. (4), where $z$ is the distance from the neutral axis of the beam.

$$w(x) = ax^2 + bx^3 + cx^4 + dx^5 + ex^6 \tag{3}$$

$$\varepsilon(x) = -z\frac{\partial^2 w}{\partial x^2} \tag{4}$$

The strain is particularly evaluated with an $x$ coordinate equal to the location of strain gauges along the tower, and the result obtained via the numerical model is compared with the stress value acquired experimentally during the decay. Comparison showed that a coefficient equal to $k = \frac{1}{1.27}$ should be applied to measured stresses in order to correct the reading of extensometer bridges as it is acquired with the acquisition system available on board of the platform. In Figure 10 the time histories of strain as measured by strain gauges and as evaluated with the modal model are reported.

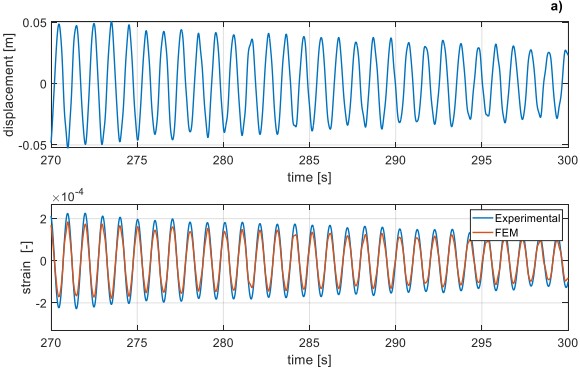

**Figure 10: Displacement and strain time histories during decay**

### 5.2.2. Blade strain gauges calibration

The strain gauges measuring flapwise deformation at blade root, placed on a specific blade and used for the analysis, were experimentally calibrated in the laboratory at Politecnico di Milano. The moment-deformation curve was obtained applying increasing weights at blade tip with the blade laying horizontally oriented in a way that the loads would cause deformation in flapwise direction only. The resulting moment-deformation expression is $M = 1.56\,\varepsilon$, where $M$ is the moment measured in Nm and $\varepsilon$ is the strain at blade root measured in μm/m.

### 5.2.3. Thrust curve

An ensemble of operating points is used to evaluate the experimental thrust curve. Firstly, the thrust force was evaluated by means of base tower strain measurements. In each operating point the mean value of strain recorded by the strain gauges is used to estimate the bending stress on tower fore-aft and side-side bending axis; then, by knowing the geometry of tower cross-section and then the distance between strain gauges location and rotor axis it is possible to evaluate the forces giving rise to the evaluated bending stress. Forces are then projected in the along-wind direction and the value of thrust is obtained. Prior of this operation, to avoid including undesired signal offsets and tower drag (even if the tower drag can be deemed negligible with respect to the amount of thrust force)



into the thrust value, the offsets of the tower base fore-aft and side-side deformation signals are evaluated. The procedure is similar to the one presented for the operating points evaluation in Sect. 4.3, considering the average values of 10-seconds windows selected under constraints that identify that the turbine rotor is parked and subject to low aerodynamic loads (inquiring rotor speed less than 5 rpm and collective blade pitch between 80 deg and 100 deg i.e. feather position). The offset applied to each signal is the one closer in time to the time of signal

acquisition.

The resulting thrust force calculated for each operating point is reported in Figure 11 as function of wind speed (dots) together with the thrust curve evaluated averaging the points as explained in Sect. 4.3 (triangles) and compared to the numerical target curve drawn in the design phase by means of FAST simulations (line).

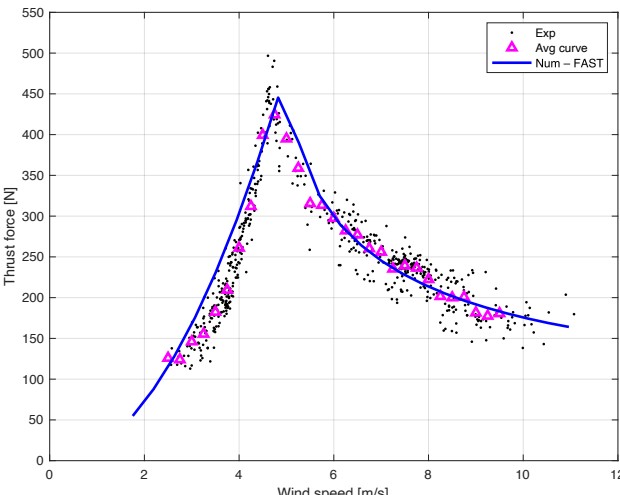

**Figure 11: Comparison between target and experimental thrust curve evaluated from tower deformation**

Secondly, the thrust force was evaluated by means of blade deformation measurements. The strain values measured at blade root for a single blade allow to calculate the flapwise moment at blade root applying the experimental moment-deformation curve. Once that the blade root flapwise bending moment is estimated, it is necessary to estimate the equivalent force giving rise to the bending moment, that is the same force generating the

rotor thrust, together with the forces acting on the other two blades. The equivalent force here mentioned is the force whose intensity is equal to the integral of aerodynamic distributed forces all along the blade axis and whose point of application is yielding the same blade-root bending moment of the whole distribution, essentially the barycenter of force distribution. The point of application of the equivalent force is estimated numerically thanks to FAST simulations performed in the same wind and operational conditions of experimental measures. The force,

projected according to the blade pitch angle, is then multiplied three times to obtain the global thrust force acting on the rotor. As an intermediate sanity check on acquired measurements, the moment at blade root experimentally measured is compared to FAST simulation results: this procedure is followed to check directly bending moment values and avoid the uncertainty given by the numerical estimation of the point of application of the equivalent, one-blade-only thrust force. The rotor thrust force estimated according to the mentioned procedure is reported in

Figure 12, with a notation equivalent to the one of Figure 11.



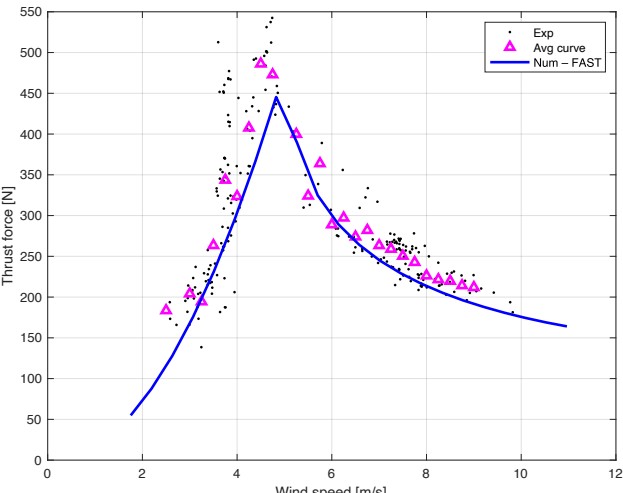

**Figure 12: Comparison between target and experimental thrust curve evaluated from blade deformation**

The thrust curves obtained with the two methods are compared to the numerical target curve in Figure 13: target curve (line), average points from tower deformation (red triangles) and average points from blade deformation
(green triangles).

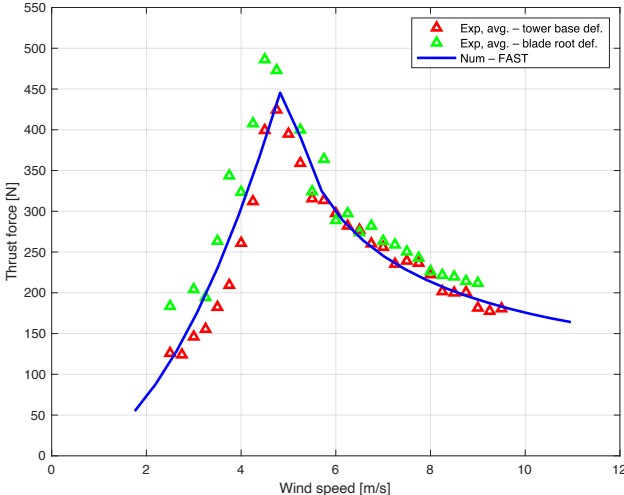

**Figure 13: Comparison between the thrust curve evaluated with the two methods presented.**

The first consideration to be made on the comparison between numerically evaluated thrust curve and the
experimental ones is that there is a good agreement between the numerical design and the large-scale physical model. The values are comparable and the trend is superimposable In this sense the aerodynamic performances of the rotor are validated also as far as rotor thrust is concerned. However, small discrepancies can be found, and



some comments on them are provided here: the curve obtained from the blade results to be higher with respect to the tower one for all wind speeds while both curves are further from the target for below-rated wind. In below-rated region, as it is stated in Sect. 6.1, some improvements have been performed on the control system to ease the startup of the wind turbine in unfavorable wind conditions; this fact causes the rotor speed found experimentally to be slightly different from the one predicted by the numerical model (see Figure 14), then causing also a difference in thrust force. Moreover, as stated in Sect. 4.2, below-rated wind speeds seems to be more affected by rotor shades effects, with an obvious shift in the thrust curve values. Concerning the discrepancy found between the curves, with the blade thrust being a bit higher than the tower-estimated one, it is to be considered that some approximation is inserted in the procedure of thrust estimation by assuming the numerical distribution of aerodynamic forces on the blade span, being impossible to obtain it on the model.

## 6. Controller verification

The correct operation of the wind turbine is assured by the control system. The power controller, during the operation of the machine, acts on the generator torque and on the pitch angle of the blades to regulate the rotor speed for the different inflow conditions. A description of the control system installed on the machine is given in (Muggiasca S. , et al., 2021) . Generator torque, blade pitch and rotor speed directly depend on the controller. Also the thrust force indirectly depends form the control action, and thus the main static and dynamic loads acting on the system. For this reason, it is important to assess that the operation of the controller during tests correctly reproduces the design target, both in terms of steady-state operating points and dynamic response.

Particular care is given to the dynamic effect of the pitch controller. In above-rated operations, the turbine is controlled varying the pitch angle and the variation of pitch angle has a direct influence on the thrust force: increasing the pitch angle of the blade the thrust force acting on the rotor decreases. This leads to a dynamic loading of the structure at a frequency determined by the control action. Moreover, as it is well known (Larsen & Hanson, 2007), instability phenomena can arise in FOWT due to the coupling between the control action, which is determining the frequency of the pitch-controlled drivetrain and thus the loading frequency of the thrust force, and the rigid body modes of the floater, in particular the pitch mode.

In order to assess the controller operation, firstly a steady-state analysis of the operating points in the whole working range of the turbine is performed. Secondly, the dynamic effect of the pitch controller action is investigated. This was performed for different gain sets in order to study the dependence on them.

### 6.1. Steady-state operating points

This paragraph reports the assessment of controller design concerning the correct reproduction of the operating points prescribed in the design phase; to this end, the design rotor steady-state performances are compared to the experimental ones. For each operating point found as in Sect. 4.3, the turbine operating parameters are collected. The result of the analysis is reported in Figure 14 where rotor speed, blade collective pitch and rotor power are represented as function of wind speed: dots represent the operating points, triangles represent the averaged values (see Sect. 4.3) and the line represent the design curves. As it can be seen there is a slight discrepancy of all the curves in the below-rated region while in the above-rated region a slight shift can be notice in blade collective pitch however preserving the correct trend. The first effect can be explained by bearing in mind that in the below-rated region some modifications has been made to the torque controller with respect to the design phase and these



modifications induced alterations in the actual operating points of the machine. The modifications were considered necessary during the setup of the experimental campaign to guarantee an easier turbine operation: since the wind turbine is started on demand, the startup has to be feasible also at wind speeds much higher than the cut-in value. As consequence, a motoring torque was introduced to enable the turbine startup and the demanded torque curve

in partial-load was adjusted to ease the startup in particular cases when the aerodynamic torque for a specific rotational speed resulted too low for the turbine to reach the correct operating point. The latter can be considered the reason of the discrepancies found. Moreover, because of the greater effect of the rotor found on the wind speed measure and despite the correction applied (see Sect. 4.2), the wind speed measure in the below-rated region presents a higher grade of uncertainty.

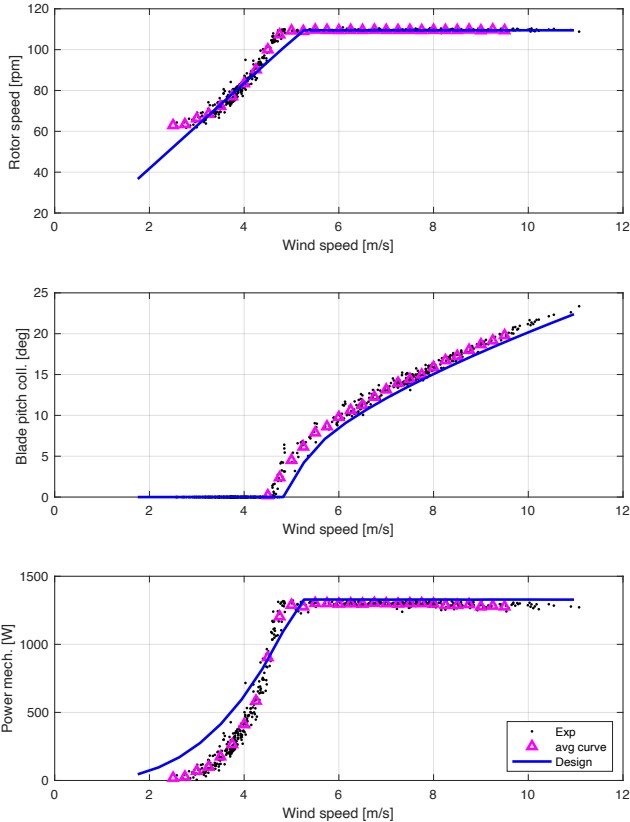


**Figure 14: Steady-state controller performances overview**

### 6.2. Pitch control effect

The effect of the pitch controller on the wind turbine dynamics is investigated evaluating the PSD of rotor speed and blade pitch for different gain sets (kp and ki) of the pitch controller (Figure 15): baseline gains are multiplied

from 0.5x to 3x to observe their influence on rotor speed oscillation and blade pitch control effort. The PSD has



been evaluated for signals of the same duration belonging to consecutive tests repeated in similar wind and waves conditions with different gain sets The rotor speed and the pitch angle respectively represent the controlled output and the control input.

On the low frequency range, between 0 and 1 Hz, it is possible to see how the gain sets affect the dynamic
amplification of wind turbulence by the transfer function of the controlled drivetrain: as the gains are increased, the peak shifts to the right and thus the pitch-controlled drivetrain becomes "stiffer". In the "3x" case the peak is highly amplified with respect to the other cases, mainly because the frequency of the controlled drivetrain is getting close to the first natural frequency of the tower (1 Hz): the thrust force, whose natural frequency of oscillation is given by the blade pitching, is forcing the tower near its first mode. Increasing the gains the amplitude
of the pitch actuation PSD increases in the whole frequency range, reflecting an increased actuation effort. However, increasing the gains does not always result in reduced rotor speed oscillations and consequently lower amplitude in the rotor speed PSD: for the "3x" case an increased fluctuation is noticed and other cases don't show a reduction with respect to the baseline as would be expected. After this investigation of controller dynamics sensitivity to gain changes the baseline gains were adopted in the rest of the experimental campaign.
From Figure 15 also the main characteristic frequencies of the rotor system can be pointed out. The identified frequencies are compared with the ones resulting from the design phase in Table 3. A good agreement was found, enhancing the quality of the design.

**Table 3: Characteristic frequencies of the rotor**

|  | Design phase [Hz] | Experimental [Hz] |
|---|---|---|
| Drivetrain | 0.39 | 0.2 – 0.8 |
| 1P | 1.82 | 1.8 |
| 3P | 5.47 | 5.5 |
| Blade-hub 1st | 7.08 | 7.2 |


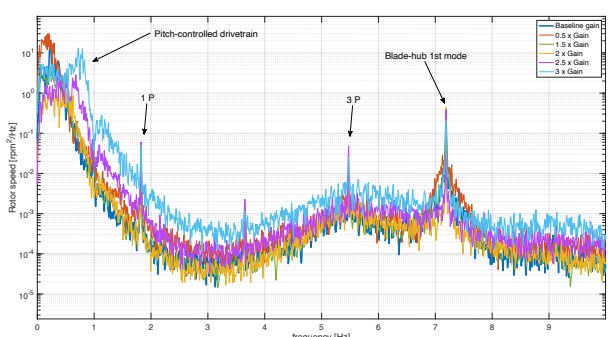

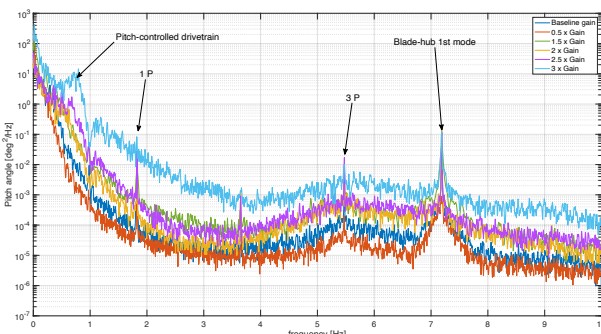

**Figure 15: PSD of rotor speed (top) and blade pitch (bottom)**

### 7. Conclusions

This work focussed on the experimental validation of the design of a large-scale wind turbine model installed on a floating multipurpose platform. The aim of the validation is to assess that the behaviour of the wind turbine model is respecting the parameters established during the design phase in terms of structural dynamics, aerodynamics of the rotor and power controller dynamics. The evaluation of structural dynamics revealed some discrepancies due to assumptions made in the numerical model of the tower in terms on slight mismatch on flexible natural frequencies; nevertheless those deviations are acknowledged and taken into account in the rest of the assessment. The aerodynamic design was evaluated in terms of thrust force exerted by the rotor. On this point, great care was put because the correct reproduction of thrust force is the key point of the whole scaling process that generated the dimensions of the wind turbine model. Rotor thrust is evaluated in two ways: measuring tower base loads and measuring blade root loads. In both cases the agreement between numerically predicted and experimentally observed thrust is very good, apart from some discrepancies in below-rated operation given by deviations of the control system from the design behaviour made during the experimental campaign and uncertainty on the wind speed measurement. The controller operation revealed a very good numerical-experimental agreement in the above-rated region for rotor speed, blade pitch and rotor power. Even in these quantities deviations are observed in the below-rated region, due to the aforementioned changes made on the controller settings.

The overall outcome of the investigation is a good matching between the desired and the observed characteristics of the large-scale model that ensures on the validity of the design process. The findings exposed in this work are on one side consolidating novel procedures in the design of large-scale models and on the other side are encouraging in further attempts in the field of large-scale modelling in natural environment.

### Author contribution

Federico Taruffi, Simone Di Carlo and Sara Muggiasca designed the tests. Federico Taruffi and Simone Di Carlo carried out the tests, performed the analyses and prepared the manuscript with alle the contributions. Sara Muggiasca supervised the analyses and reviewed the manuscript. Marco Belloli is responsible for supervision, funding acquisition and project administration.



**Competing interests.** The authors declare that they have no conflict of interest.

**Aknowledgements**

This work has been produced in the framework of the Blue Growth Farm project (http://www.thebluegrowthfarm.eu/), which has received funding from the European Union's Horizon 2020 research and innovation programme under Grant Agreement number 774426. The content of the work does not report the opinion of the European Commission and reflects only the views of the author(s), including errors or
omissions. The European Commission is also not liable for any use that may be made of the information contained herein.

**Data availability.** The dataset is accessible upon request to the authors.

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
