# Peer review of "A large-scale wind turbine model installed on a floating structure: experimental validation of the numerical design"

_Wind Energy Science, 2022_

## Author Response (AR1)

**Answer to Reviewers**

Dear referees,
thank you for your comments. All the concerns that were pointed out have been accomplished or clarified. Please find below the point-by-point response for each referee.

Answer to Referee #1:

The paper is overall interesting and well organized. Please try to improve it following the comments below:

1. The authors need to cite paper doi:10.1088/1742-6596/2265/4/042008, also authored by them. The paper has some portions which are very similar to those of the one under consideration. Some figures are also identical. The authors need to clearly state which content is novel and which is not. As a general recommendation, results need to be repeated only when they bring added value to the new publication in terms of clarity or completeness. Too similar figures should be avoided.

    The citation to the referred paper is added in the introduction, specifying that the part of the analysis concerning the thrust curve is completely new, and the methodology adopted even to process already presented data is improved and specified in the body of the paper. Figure 9, 14 and 15 are reporting parameters already analysed in a previous work, but the processing of data behind the figures is improved. First of all, more operative points (i.e. more data) are taken into account in the present work; then, the pre-processing of data is improved to more effectively remove non-significant entries.

    *Changes in line 58 (marked-up manuscript): "A preliminary assessment of the wind turbine aerodynamic design is reported synthetically also in (Taruffi, et al. 2022), with a focus on the methodology adopted to process data. In this work the methodology used to perform the analysis is improved and detailed. Results are showed in a more extensive way and some of the wind turbine properties, like the thrust curve, are here presented for the first time."*

2. Please state which version of FAST (or OpenFAST?) has been used.

    The version is FAST v8.16.

3. Uncertainty estimation on experimental data could bring some added value.

    The uncertainty estimation on experimental data is more complicated in the context of this tests, given the natural environment and the consequent uncontrollability of the inputs (e.g. wind speed and direction). The single components and/or sensors were individually verified but given the timing of the site and the characteristics of the installation itself, it was not possible to make an accurate on-site assessment of the entire measurement chain. In future analyses the authors will try to evaluate the repeatability of the measurements and the variability as the input varies. However, graphs including dispersion were added in the manuscript (Figure 11 and 12) regarding the evaluation of thrust force.

4. It seems that the authors did not use a yaw control (is this the case?). If so, I would suggest plotting the wind direction trend vs time to demonstrate to the reader that yawed functioning conditions are not affecting the results.

    The yaw control wasn't activated during the tests, mainly because of concerns on the wind direction measurements, on which the controller relies, that shows fluctuations during transients. The yaw was aligned to the wind direction before each startup of the turbine.

However, if the wind direction changes during the tests the rotor faces a misaligned wind that can alter the performance. For this reason, the misaligned cases are excluded in the analysis: in the evaluation of the regime points the points that present a wind misalignment outside a threshold of 5 deg are discarded.

*Changes in line 164 and 263 (marked-up manuscript): "The yaw control is not activated. […] Since the yaw control is not active during tests, the alignment of the rotor with the wind direction is checked too, discarding frames with greater misalignment that can show performance altered with respect to the ideal, fully aligned case."*

5. The English form has some minor flaws. The most apparent one is "performances" that should be singular in technical writing. A revision by a native English speaker is suggested.

   Mistakes are corrected in the text.

6. The explanation for the discrepancies seen in Figure 14 is a little bit vague. Please try to better motivate the loss of performance in the below-rated region. Could this be associated with aerodynamics?

   The discrepancies between the design curves and the measurements observed can be motivated with two main causes. The first apply for the below-rated region only, where the discrepancies are higher, and is related to the modification of the demand curve of the torque controller. This modification was introduced during the setup tests performed on-site and it was introduced to overcome problems on the startup of the turbine, that was found to be problematic in specific cases and in particular for high wind speeds. The modification consists of the introduction of a motoring torque for low rotational speed and on the shift of the speed thresholds. This directly leads to the modification of the operating points (rotor speed and torque) in the region. The modified curve can be seen in doi:10.1088/1742-6596/2265/4/042008. The second reason is related to the uncertainty that persist in the wind speed measurements also after the correction is applied, also given the complexity of the procedure. This can slightly affect the measurements both in below- and above-rated region. However, also a better-than-expected efficiency of the profile could explain the slight upward shift in the pitch angle curve in above-rated. The latter could also be explained by the lower above-rated torque (and thus power) found in above-rated and derived by the oscillation of the rotor speed causing a non-constant torque demand from the controller that results, in average, lower than the maximum torque. This would cause the increase of the pitch angle that is seen. The whole explanation is also summarized in the text.

   *Changes in line 440 (marked-up manuscript): "Moreover, despite the correction applied (see Sect. 4.2) a grade of uncertainty persists in the wind speed measure and this can affect the curves both in below- and above-rated regions resulting in a slight horizontal shift. However, also a better-than-expected efficiency of the blade profile could explain the slight upward shift in the pitch angle curve seen in above-rated."*

7. As a general impression, the paper is very "descriptive", i.e., shows good results, but probably lacks a little bit of a critical perspective. I would recommend revising the final part of the paper in a way that could provide to the reader a more critical insight for example on the validity of the methods used for future studies, on the source of uncertainties in the real environment or in some prescriptions to manage wind turbines in multi-purpose platforms.

   Considerations about procedures for tower dynamics inspection, thrust estimation, and controller adaptation are added in the conclusions (changes in all section 7).

Answer to Referee #2:

The manuscript provides an experimental validation of the design of a large-scale wind turbine model installed on a floating multipurpose platform. The paper is clear, relevant, and scientifically sound. Some questions/comments are added below to further improve the quality of the manuscript.

1. The paper could say a few more words on the overall advantages/disadvantages of the multiplatform concept. For example, are there any studies that showed the potential benefits of combining floating wind and wave energy converters from a system point of view? And also with aquaculture?

   A reference about the economical convenience of multi-purpose platform is added, with a focus on mooring systems and electrical dispatch system sharing. Two references about studies on favourable effects on combined power generation from wind and waves is added. No reference was found on fish farming dedicated multipurpose platforms.

   *Changes in line 25 (marked-up manuscript): "A multipurpose platform is a floating platform hosting different technologies for contemporary energy and food production. In this way different activities can improve their redditivity by sharing common and expensive facilities, like the platform itself, the mooring system, the electrical dispatch system (Aubault 2011) and so on. (Michailides 2014) and (Muliawan, Karimirad and Moan 2013) investigated the system dynamics and power generation of multipurpose platforms for wave and wind energy production, revealing an improvement of combined power generation. A partly different example"*

2. Line 39: The paper mentions that wave-tank tests have been performed at ECN. Can the authors comment on how the present 1:15 model scale results compared to the small- scale results? This would help support the statement that "large-scale models are technologically very similar to prototypes and can reduce scaling effects". Alternatively, the authors can point to a reference showing this.

   In Ruzzo et al. "On the arrangement of two experimental activities on a novel multipurpose floating structure concept" is shown the comparison between the two experimental setup. The reference is cited in the text

3. Page 3 Line 82: the choice of the airfoil is justified by the fact that it is suitable for low Reynolds number applications. However, the environment here is more realistic (i.e. higher Re) than in a wind-tunnel environment. Can you give the typical Re values encountered here and explain why the low-Re airfoil was still a good choice?

   Even though the environment of this application is realistic, the model faces a Reynolds number mismatch of around 60 with respect to the full-scale. This is way less if compared with wind tunnel values but justifies the choice of the profile. The profile thickness results an intermediate choice between the full-scale and a wind tunnel model. More information, including the analogy between the selection of the profile for outdoor and wind tunnel applications and values of the Reynolds number along the blade span, can be read in doi:10.3390/en14082119 (Figure 3). This reference is cited in the text.

   *Changes in line 89 (marked-up manuscript): "For the blade airfoil the SG6040, an intermediate choice between the full-scale and a typical wind tunnel application, was selected and experimentally characterized by means of wind tunnel tests on a 2D model."*

4. Page 4 L 122. The goal is to investigate the effect of floating motions on the turbine power production and structural health. However, because the platform is quite large compared to

conventional floating wind turbines, the motions are expected to have a much smaller impact on the turbine. Can the authors comment on how the floating motions of this multiplatform differ from those of a floating wind turbine without aquaculture/WEC? Also, are scaling effects, i.e. between full scale and a 1:15 model, expected to be larger/smaller/identical for a floating wind turbine when the floating platform is smaller?

The necessary premise to be done here is that the platform dynamics is still to be investigated properly. What can be said is that the platform built during this project is surely bigger (in relative terms) than a conventional floating wind platform, and then in general terms less prone to intense motions. However, by inspecting the natural periods of the platform, it is found that pitch and roll motions are located in a frequency range where also longer waves of the first order spectrum are acting. This is triggering time by time a dynamic amplification effect on these motions, that was clearly visible during the experimental campaign. It is to be understood if a full-scale platform in a full-scale sea would suffer from the same issue in a more or less relevant way. Concerning the scaling effects, the most critical issue to be handled during the scaling procedure, that is not present in traditional floaters, is the moonpool effect that is dependent from viscous phenomena.

5. Pages 7-8. Can the authors provide more information on the post-processing, i.e. filter used, decrement procedure? Alternatively, the scripts could be openly shared so that the procedures can be reproduced.

   Specification about the adopted filter is added in the manuscript. A reference about the logarithmic decrement procedure is added, together with the specification of assumptions that stand behind this procedure.

   *Changes in line 190 (marked-up manuscript): "To remove these undesired frequency components a bandpass filter is applied to the signal; in detail a 6th order Butterworth filter with lower and higher cut-off frequencies respectively of 0.6 and 5 Hz. The application of the filter allows to obtain the 1-dof decay corresponding to the first mode of vibration (Figure 7). Analysis of the signal revealed a first mode frequency of 0.98 Hz. As far as damping is concerned, by assuming a 1-dof linear behaviour of the system the logarithmic decrement procedure is applied (Cheli and Diana 2015), and a damping ratio of 0.33 % is found."*

6. Fig 15: What is the peak at around 3.7Hz?

   The peak is at the double of the 1P frequency (1P is 1.8Hz for above rated operating conditions).

7. Line 348: Once that --> Once

   It is corrected in the text.

8. Line 371 & 427: punctuation missing

   It is corrected in the text.